# Characterization of Polyvinyl Alcohol (PVA)/Polyacrylic Acid (PAA) Composite Film-Forming Solutions and Resulting Films as Affected by Beeswax Content

**DOI:** 10.3390/polym16030310

**Published:** 2024-01-23

**Authors:** Woo Su Lim, Min Ha Kim, Hyun Jin Park, Min Hyeock Lee

**Affiliations:** 1Department of Biotechnology, College of Life Science and Biotechnology, Korea University, Seoul 02841, Republic of Korea; 2Department of Food Science and Biotechnology, College of Life Sciences, Kyung Hee University, Yongin 17104, Republic of Korea

**Keywords:** polyvinyl alcohol, beeswax, emulsion stability, structural properties, barrier properties

## Abstract

Recently, the food packaging industry has focused on developing an eco-friendly and sustainable food packaging system. This study describes the effect of beeswax on the physical, structural, and barrier properties of a polyvinyl alcohol (PVA)/polyacrylic acid (PAA) composite film. The incorporation of beeswax improved the barrier properties against oxygen, water, and oil. However, the addition of a high content of beeswax caused phase separation in the film-forming solution. The destabilization mechanisms such as clarification and creaming formation in the film-forming solution were revealed by turbidimetric analysis. The results of scanning electron microscopy (SEM) and confocal laser scanning microscopy (CLSM) indicates that non-homogeneous structures in the film-forming solution were formed as a function of increased beeswax content due to the agglomeration of beeswax. The mechanical properties of the films were also evaluated to determine the most appropriate content of beeswax. There was a slight decrease in tensile strength and an increase in elongation as beeswax content increased up to 10%. Thus, the PVA/PAA composite film with 10% beeswax was chosen for further applications. In summary, the PVA/PAA composite film developed in this study with 10% beeswax exhibited a significant improvement in barrier properties and has the potential for use in commerce.

## 1. Introduction

Food packaging materials serve to safeguard food products by shielding them from external environmental factors. The mechanical and barrier properties of these materials are crucial in ensuring the protection of food items. Consequently, plastics have been widely regarded as an appealing choice within the food packaging industry [1]. Petroleum-based plastics like polystyrene (PS), polyvinyl chloride (PVC), and polyethylene (PE) have been utilized since the mid-20th century due to their excellent mechanical and barrier properties, coupled with cost effectiveness [2]. Nevertheless, these plastics pose a significant environmental threat as they degrade poorly in natural settings, potentially resulting in severe environmental pollution [3]. In response, the food packaging sector is actively seeking to curtail the usage of petroleum-based plastics and explore alternative materials. Consequently, numerous researchers are dedicated to devising biodegradable packaging options to replace these petroleum-based polymers.

Polyvinyl alcohol (PVA) stands out as a synthetic biopolymer extensively utilized in the packaging industry. Its appeal lies in several advantages, including biodegradability, non-toxicity, and remarkable mechanical properties [4,5,6]. However, a notable drawback of PVA is its high water solubility, which poses a significant challenge when used as a food packaging material. Consequently, numerous efforts have been made to address this issue and improve the physicochemical properties of PVA through structural modifications, such as incorporating nanoclay and blending it with other polymers [7,8,9]. Acrylic acid finds diverse applications across various industries owing to its cost effectiveness. Additionally, it boasts ease of synthesis and inherent characteristics of high water solubility and biodegradability [10]. Polyacrylic acid (PAA) is extensively employed as a cross-linking agent in the preparation of PVA films due to its multiple carboxyl groups, facilitating crosslinking with the hydroxyl group of PVA by forming an ester bond [11]. Studies have indicated that alterations in the polymer structure resulting from crosslinking can impact both barrier and physical properties [12]. However, the use of PVA/PAA composite films in food packaging is limited due to their high water vapor permeability (WVP). Consequently, further research is necessary to enhance the moisture barrier properties of these hydrophilic polymers.

To overcome this limitation, lipophilic materials such as fat, oil, and wax have been considered to improve the water barrier properties of hydrophilic polymer films. Among these, wax, due to its high hydrophobicity owing to long-chain fatty alcohols and alkanes, stands out as an excellent moisture blocker [13]. Natural waxes, sourced from insects and plants, are preferred over synthetic ones due to their renewable nature [14], aligning with the growing consumer preference for natural substances [15]. As a result, considerable research has been dedicated to enhancing the barrier properties of films by applying wax [16]. In this study, beeswax, derived from honeybee wax glands, was incorporated into PVA/PAA composite films due to its chemical components consisting of esters, free fatty acids, and hydrocarbons, primarily palmitate, palmitoleate, and oleate esters of long-chain alcohols [17].

Surfactants play a crucial role in stabilizing emulsions, yet their excessive use in industrial applications can increase film moisture sensitivity and reduce gloss [18]. Thus, minimizing surfactant content in the emulsion film is essential. However, insufficient surfactant can lead to phase separation during film preparation through destabilization mechanisms like flocculation, coalescence, and creaming formation [19,20,21]. Emulsion polymerization involves emulsifying hydrophobic materials with a monomer that acts as an oil-in-water emulsifier, controlling droplet size distribution by adjusting monomer or emulsifier concentration or type [22]. High shear processes can reduce polymer particle sizes, serving as nanoreactors or microreactors [23,24]. The emulsifier used in this study was PVA, which is a stabilizer in emulsion polymerization and forms an emulsion without other surfactants. In addition, PVA can act as a film matrix and form composite films with PAA. The incorporation of hydrophobic compounds into the polymer matrix effectively enhances water barrier properties but can impact film physical, mechanical, and morphological properties. However, improper use of hydrophobic compounds in emulsion films may lead to defects and property failures.

Therefore, this study aims to assess the impact of beeswax content on the emulsion stability of the film-forming solution and its influence on the structural, mechanical, and barrier properties of PVA/PAA composite films. Structural property investigations of these films with beeswax were conducted using Fourier transform infrared (FTIR) measurements and X-ray diffraction (XRD) analysis. The emulsion dispersion stability and destabilization mechanisms of the film-forming solution during storage were determined by using turbidimetric assays. Additionally, the study investigated film properties, including water vapor, oxygen, and oil barrier properties, as well as mechanical properties.

## 2. Materials and Methods

### 2.1. Materials

PVA (Mw = 22,000 Da, CAS Number 9002-89-5) was purchased from Junsei Chemical Co. (Tokyo, Japan). White beeswax (CAS Number 8012-89-3) was purchased from Samchun Pure Chemical Co. (Pyeongtaek, Republic of Korea). The saponification and acid values of the beeswax are 87–104 and 5–22, respectively, and the melting temperature of the beeswax is about 60.0–67.0 °C. PAA (Mw = 5000 Da, CAS Number 9003-01-4) was purchased from FUJIFILM Wako Pure Chemical Co. (Tokyo, Japan). Distilled water (extra pure grade) used in this study was purchased from Duksan Co. (Ansan, Republic of Korea).

### 2.2. Film Preparation

#### 2.2.1. Film-Forming Solution

PAA (1%, *w*/*v*) was dissolved in distilled water. After the PAA was sufficiently dispersed, PVA (4%, *w*/*v*) was added and heated to 80 °C. Subsequently, different amounts of beeswax were added to the primary film-forming solution (final concentrations; 0%, 1%, 5%, 10%, 15%, and 20%; *w*/*w* total solid contents) and stirred for 4 h. The suspensions were homogenized to 80–85 °C for 5 min at 13,000 rpm using a high-speed homogenizer (Ultra-Turrax T18, IKA, Staufen, Germany). The degassing process was carried out for 5 min in an ultrasonic bath (Branson 2510, Branson Ultrasonic Corp., Brookfield, CT, USA).

#### 2.2.2. PVA/PAA Composite Film

Prepared film-forming solutions (15 mL) was poured into a Petri dish with an inner diameter of 90 mm. All film samples were dried in a thermo-hygrostat (Model TR-001-1, JeioTech Co., Ltd., Daejeon, Republic of Korea) at 25 °C with 50% relative humidity for 48 h.

### 2.3. Emulsion Stability

#### 2.3.1. Centrifugal Stability

The centrifugal stability of the film-forming solution was obtained to investigate creaming of beeswax. The creaming index was determined using a previously reported method with minor modifications [25]. The film-forming solutions (50 g) were centrifuged at 3134× *g* for 10 min (ScanSpeed 1248R centrifuge, Labogene, Lillerød, Denmark). The film-forming solutions were filtered through filter paper. The centrifuge tubes were rinsed twice with distilled water (50 mL each) and filtered. The filter paper was dried in an oven at 80 °C for 24 h and the weight change of the filter paper was measured. The emulsion stability was calculated as follows:(1)$$Creaming\;index(\%)=\frac{Weight\;change\;of\;the\;filter\;paper}{Amount\;of\;beeswax\;used\;for\;formulation}\times 100$$

#### 2.3.2. Laser Scanning Turbidimetry

The phase separation or dispersion stability of the film-forming solution was determined by using a Turbiscan ASG (Formulaction, L’Union, France) equipped with a source of infrared light of λ = 880 nm. Each sample (15 mL) was carefully pipetted into a transparent glass vial and placed in a Turbiscan tower. The measurement performed for 1 h after stabilization for 5 min at 25 °C. Each sample was checked for dispersion stability using backscattering (BS%) profiles at various heights from bottom to top at 40 μm intervals. Turbiscan stability index (TSI) was used to evaluate the dispersion stability of beeswax in the film-forming solution. The TSI was calculated automatically with the Turbiscan software (version 1.3) as follows:(2)$$TSI=\sum_{i}\frac{\sum_{h}\left|{scan}_{i}\left(h\right)-{scan}_{i-1}(h)\right|}{H}$$
where *H* is the sample height from the bottom of the glass cell to the meniscus, *i* is the time from 1 to *k* (*k* = total scanning time/scanning speed), *scan_i_* (*h*) is the average backscattering for each time (*i*) of measurement, and *scan_i−_*_1_ (*h*) is the average backscattering for the *i*−1 time of measurement.

### 2.4. Physical Property of the Film

#### 2.4.1. Fourier Transform Infrared (FTIR)

FTIR measurements of the films were performed using an Agilent Cary 630 FTIR spectrometer (Agilent Technologies, Danbury, CT, USA) in attenuated reflectance (ATR) mode. The FTIR spectra were recorded in the range of 4000–600 cm^−1^. The spectra were collected in scans at a resolution of 4 cm^−1^ for each film. All testing films were conditioned in a constant state at 25 °C and 50% relative humidity.

#### 2.4.2. X-ray Diffraction (XRD) Analysis

The crystallinity of the films was analyzed using XRD (X’PERT MPD diffractometer, Philips, The Netherlands) at a rate of 2° per min (range, 10–30 °) using a current of 30 mA and voltage of 40 kV. All testing films were conditioned in a constant state at 25 °C and 50% relative humidity.

#### 2.4.3. Surface and Cross-Section Microstructure

The micro morphologies of the surfaces and cross-sectional areas of the films were observed by field emission-scanning electron microscopy (FE-SEM) (Hitachi S-4800, Hitachi, Tokyo, Japan). All the samples were fixed with carbon tape coated with a thin layer of platinum under vacuum for 2 min. All testing films were conditioned in a constant state at 25 °C and 50% relative humidity.

#### 2.4.4. Confocal Laser Scanning Microscopy (CLSM)

The morphological structure of the beeswax incorporated film was visualized by confocal laser scanning microscopy (CLSM) (LSM 700, Carl Zeiss, Oberkochen, Germany). The film was stained with Nile Red solution (1 mg mL^−1^ in isopropyl alcohol) for lipid marking. The film specimen (10 mm × 10 mm) was mounted on a glass slide and coated with a cover glass using glycerol. Beeswax droplets dyed in Nile Red were excited red at 488 nm. CLSM images were analyzed using the ZEN 2009 software (Carl Zeiss MicroImaging, Jena, Germany). All testing films were conditioned in a constant state at 25 °C and 50% relative humidity.

#### 2.4.5. Water Absorption and Solubility

The water absorption and water solubility of the films were determined via gravimetric analysis [15]. The film was cut into 20 mm × 20 mm pieces and weighed (*M*_1_). Film samples were completely dried at 105 °C for 24 h (OF-21, JeioTech Co., Ltd., Daejeon, Republic of Korea) and weighed (*M*_2_). The dried film was immersed in 30 mL of distilled water and maintained at 25 °C for 24 h. The residue film sample was removed, dried at 105 °C for 24 h, and weighed (*M*_3_). The moisture content and solubility of the film were calculated as follows:(3)$$Moisture\;content\;(\%)=\frac{M_{2}-M_{1}}{M_{1}}\times 100$$
(4)$$Solubility\;(\%)=\frac{M_{1}-M_{3}}{M_{1}}\times 100$$

### 2.5. Barrier Property of the Film

#### 2.5.1. Water Vapor Permeability (WVP)

The water vapor permeability of each film was measured using a cup method according to the American Society of Testing and Materials (ASTM) standard (E96-95) [12]. Distilled water (10 mL) was placed in the test cup, and the film sample was completely covered with the O-ring. Cups were incubated at 25 °C in a thermo-hygrostat chamber with 50% relative humidity for 12 h. The calculated results were used to measure the water vapor permeability using the following equation:(5)$$WVP=\frac{∆m\times d}{∆t\times S\times ∆P}$$
where Δ*m* (g) is the cup mass variation measured after a period of 12 h, *d* (m) and *S* (m^2^) indicate the thickness and area of the films, respectively, Δ*t* (s) represents the time intervals, and Δ*P* (Pa) represents the pressure difference (1753.55 Pa and 133.32 Pa) between both sides of the films.

#### 2.5.2. Oxygen Permeability

Oxygen permeability was evaluated using an OX-TRAN^®^ Model 2/61 apparatus (MOCON Inc., Brooklyn Park, MN, USA) according to ASTM standard (D3985) at 23 ± 1 °C and relative humidity of 0% [12]. Gas flow rate was fixed at 10 mL/min. The oxygen permeability is reported as cc·m/m^2^·day at 101.325 kPa.

#### 2.5.3. Oil Permeability

Oil permeability of the film was determined using the modified cup method. A film sample and five layers of filter paper were placed and fixed at the entrance of the cup loaded with 10 mL of medium-chain triglyceride oil. The prepared cups were placed upside down at 25 °C and 50% relative humidity, and the weight change of the filter paper was measured after 7 days. The oil permeability was calculated using the following equation:(6)$$Oil\;permeability=\frac{∆m\times d}{t\times S}$$
where Δ*m* (g) is the weight difference of filter papers, *t* (day) represents the testing time, and *d* (mm) and *S* (m^2^) indicate the thickness and area of the films, respectively.

### 2.6. Mechanical Property of the Film

The tensile strength and elongation at break of the film were determined using the modified ASTM standard (D889) using an Instron 3366 universal testing machine (Instron, Norwood, MA, USA) [15]. All sample films were cut to a size of 25 mm × 70 mm. Each side of the film was fixed with Instron grips, and the film was stretched at a rate of 10 mm/min. All experiments were performed using three samples. All testing films were conditioned in a constant state at 25 °C and 50% relative humidity.

### 2.7. Statistical Analysis

The data recorded are presented as the mean ± standard deviation for each experiment. Statistical analyses were conducted using Statistical Package for the Social Sciences (SPSS) software package (Version 20.0, SPSS Inc., Chicago, IL, USA). Significant differences between samples were determined by a one-way analysis of variance, followed by Duncan’s multiple range test. The results were considered significant if *p* was <0.05.

## 3. Results and Discussion

### 3.1. Emulsion Stability

Creaming layers could be formed in the emulsion systems due to the aggregation of insoluble particles, resulting from either flocculation or coalescence. Creaming serves as an indicator of emulsion instability; hence, emulsion stability was assessed by determining the creaming index under the influence of centrifugal force and gravity [26]. This index reflects the extent of agglomeration among the beeswax droplets, which are non-soluble particles filtered out during the assessment. As depicted in Figure 1, the creaming index of the film-forming solutions demonstrated an increasing trend corresponding to higher beeswax content. Notably, no observable creaming occurred in film-forming solutions containing 0% and 1% beeswax. However, a significant increment of the creaming index was distinctly noted in solutions with beeswax content exceeding 10%, indicating PVA’s failure to function as an emulsifier.

The dispersion stability of the prepared film-forming solutions was also evaluated through backscattering profiles (Figure 2), displaying backscattering values as a function of height within measurement cells containing each sample. Differences in backscattering values from bottom to top signify the dispersion’s destabilization during storage. Specifically, unstirred film-forming solutions at room temperature exhibited separation into distinct layers. This agglomeration of beeswax within the solution resulted from hydrophobic interactions with van der Waals forces [27]. Over time, increased storage led to flocculation and coalescence among the beeswax droplets within the emulsion system. Consequently, agglomerated beeswax droplets, having lower density than the aqueous phase, ascended to the top layer, causing an elevation in the backscattering value. Notably, the thickest creaming layer, caused by agglomerated beeswax droplets, was observed in film-forming solutions with 20% beeswax content. Concurrently, a reduction in backscattering was observed at the bottom layer, indicating clarification of the dispersion due to lower beeswax content [19,20,28,29]. Distinct variations in backscattering values at the top and bottom layers during storage were particularly evident in film-forming solutions with higher beeswax content (15% and 20%), indicating compromised quality characterized by irregular emulsion film structures due to phase separation during film drying. The TSI values for the samples over time are depicted in Figure 3. The TSI quantifies destabilization within the dispersion during storage, with a value closer to zero indicating higher emulsion stability [19,20]. Film-forming solutions with lower beeswax content exhibited relatively low TSI values during storage. However, TSI values increased proportionally with higher beeswax content in the solutions, showing further escalation over time. On the other hand, after 45 min, the film-forming solution with 15% beeswax exhibited a higher TSI value compared to the film-forming solution with 20% beeswax. This is attributed to the rapid coalescence of emulsion particles and the swift formation of a cream layer in the upper portion in the case of the film-forming solution with 20% beeswax during the initial period (before 45 min). In cases where rapid destabilization occurs in the initial stages of emulsion samples, the subsequent rate of change tends to decrease, leading to a slower increase in TSI values. However, both film-forming solutions containing 15% and 20% beeswax notably displayed a rapid increase in TSI values, surpassing 1 after 45 min, indicating unstable emulsion systems caused by flocculation and coalescence during storage.

### 3.2. FTIR

Figure 4 displays the FTIR spectra of PVA/PAA composite films containing varying beeswax concentrations. The FTIR spectra of individual film-forming components, namely PVA, PAA, and beeswax, are presented in Appendix A. Previous studies have noted changes in the FTIR spectra, indicating interaction between PVA and PAA [13]. This interaction involves the hydroxyl group in PVA and the carboxylic acid group in PAA, resulting in the formation of a crosslinked network through esterification. The resulting PVA/PAA composite film manifests absorption peaks at 1709 cm^−1^, attributed to the C=O stretching vibrations (1720–1706 cm^−1^) of the ester group formed via esterification [30]. Thus, the FTIR spectra indicate alterations in the chemical structure owing to the esterification reaction in the PVA/PAA composite film. Beeswax is a complex organic mixture composed of diverse components such as hydrocarbons, free fatty acids, esters of fatty acids and fatty alcohols, diesters, and exogenous substances [31]. Figure 4 reveals changes in the FTIR peaks of PVA/PAA composite films corresponding to increasing beeswax content. Notably, the FTIR spectrum of the PVA/PAA composite film with 1% beeswax closely resembled that of the control film (without beeswax). An additional peak at 1172 cm^−1^, indicating C=O stretching and C-H bending vibrations, was detected. However, the intensities of two peaks at 2921 cm^−1^ (CH_2_ asymmetric stretching vibration) and 2852 cm^−1^ (CH_2_ symmetric stretching vibration) noticeably increased with beeswax content ranging from 5% to 20%. These peaks are characteristic analytical signals of beeswax commonly observed in IR spectra [32]. Despite the presence of peaks at 1739 cm^−1^ and 1714 cm^−1^ in the FTIR spectrum of beeswax, these were not discernible in the FTIR spectra of PVA/PAA composite films due to the dominance of PVA/PAA peaks.

### 3.3. Crystallinity of the Film

XRD analysis primarily investigates the crystalline properties of the film surface, as depicted in Figure 5. Across all films, a prominent peak was observed at approximately 2θ = 19.6°, signifying a semi-crystalline PVA structure within the film [33]. Notably, the introduction of 5% beeswax to the PVA/PAA composite film did not notably alter this peak. However, in films incorporating more than 5% beeswax, the peak intensity decreased with escalating beeswax content. This implies that the film retained its crystalline nature until the addition of 5% beeswax, after which some beeswax emerged on the film surface upon inclusion of 10% beeswax. Additionally, two distinct peaks at 2θ = 21.3° and 23.6°, characteristic of beeswax XRD, were absent in films lacking beeswax or containing 1% beeswax. However, these peaks emerged in the film with 5% beeswax and steadily increased with higher beeswax content. These XRD peaks correspond to the orthorhombic structure of hydrocarbon-monoester fractions present in beeswax [34]. These XRD findings align with the earlier FTIR results (Figure 3), indicating the distribution of hydrophobic constituents from beeswax predominantly on the film surface, particularly in films with more than 5% beeswax. This suggests a lack of sufficient entrapment of beeswax within the composite film matrix by PVA. Such surface distribution of hydrophobic constituents might influence the physical and barrier properties of the films [33].

### 3.4. Surface and Cross-Section Microstructure of the Film

Figure 6 exhibits surface and cross-sectional SEM images of the film, revealing a discernible impact of beeswax content on the film’s microstructure. The PVA/PAA composite film initially displayed a relatively smooth surface structure. However, films incorporating lipophilic components exhibited irregular particles within the film matrix. Notably, PVA/PAA composite films containing more than 5% beeswax displayed rough and uneven surface structures. Additionally, pores were evident on the film surfaces, attributed to lipid evaporation. The SEM observation subjected the film specimens to high vacuum, causing the breakdown of beeswax droplets, potentially resulting in pore-like structures on the film surface [35]. Furthermore, these structures could arise from creaming formation due to the upward movement of hydrophobic droplets during film drying. Specifically, the cross-sectional images of the PVA/PAA composite film with 20% beeswax revealed numerous irregularly shaped particles predominantly situated on the upper side of the film matrix. Excessive wax incorporation or inadequate surfactant addition in the film formulation might contribute to diminished emulsion stability within the film matrix. Consequently, irregularities and roughness were observed in the surface and cross-sectional structures of the emulsion film. The formation of heterogeneous film structures, resulting from the amalgamation of wax and oil, has been previously documented in various studies [16,36]. Furthermore, considering the drying temperature (25 °C) lower than the melting temperature of beeswax (62–64 °C), accelerated wax agglomeration might account for the development of protruding wax structures within the film.

### 3.5. CLSM Observation

As previously mentioned, beeswax retains a solid-state form at room temperature, with a melting point between 62–64 °C. Consequently, during storage at the film drying temperature, beeswax tends to agglomerate, potentially leading to the formation of substandard films, such as creaming. Thus, it becomes imperative to employ CLSM to ascertain the effectiveness of the PVA/PAA composite matrix in regulating the beeswax dispersion morphology. The CLSM image in Figure 7 showcases the microstructural characteristics of the emulsion film, elucidating the dispersion of red-dyed droplets. The utilization of Nile red staining enabled the visualization of dispersed beeswax droplets within the film matrix. PVA molecules possess the capability to encapsulate and stabilize hydrophobic materials like oil and wax within the film matrix [37]. The CLSM image underscores that the dispersion stability of wax particles incorporated into the PVA/PAA composite film matrix was upheld, validating the efficacy of the PVA-stabilized emulsion in facilitating beeswax dispersion [35]. In addition, the morphological architecture of the emulsion film was highly dependent on the beeswax content. Higher beeswax content corresponded to increased particle size and greater droplet numbers. Specifically, elevated wax content resulted in irregularly shaped beeswax particle distribution, diminishing compatibility with the film matrix. This observation suggests that high levels of beeswax led to unstable emulsions and poor emulsion-forming capacity, adversely impacting film quality.

### 3.6. Water Absorption and Solubility of the Film

Table 1 presents the water absorption and solubility data for the PVA/PAA composite films containing varying beeswax concentrations. These metrics serve as crucial indicators for assessing the films’ water affinity or resistance. Notably, as the beeswax content increased, both water absorption and solubility of the PVA/PAA composite films decreased, ranging from 11.91 ± 0.11% to 8.98 ± 0.61% and from 23.98 ± 0.23% to 7.90 ± 0.03%, respectively. The introduction of hydrophobic molecules like beeswax into the film matrix can engage in hydrogen bonding interactions with polymer molecules, ultimately reducing the film’s interaction with water molecules [15]. Additionally, beeswax contributes to imparting hydrophobicity to the film surfaces, consequently diminishing water absorption and solubility. Previous research by Zhang et al. [17] supports these findings, noting that the inclusion of wax in biopolymer films aids in preserving film integrity when immersed in water. Therefore, the integration of hydrophobic compounds into the film matrix markedly decreases the film’s solubility, showcasing enhanced water resistance and reduced water absorption.

### 3.7. Barrier Properties of the Film

WVP stands as a pivotal attribute for practical applications in food packaging films. As delineated in Table 1, the WVP of PVA/PAA composite films demonstrated a decreasing trend with escalating beeswax content. Research suggests that wax, owing to its high hydrophobicity due to the abundance of long-chain fatty alcohols and alkanes, efficiently reduces WVP [38]. Consequently, the inclusion of beeswax in the film composition provides exemplary moisture barrier properties. Additionally, the distribution of beeswax, consisting of small particle sizes within the film matrix, constructs a convoluted path for water molecule diffusion, thereby extending their travel distance through the film matrix [15]. Nevertheless, in this study, there was no noteworthy disparity between the WVPs of PVA/PAA composite films containing 15% and 20% beeswax contents. This lack of distinction is attributed to insufficient PVA acting as a surfactant for emulsifying beeswax within the film matrix. Consequently, an increased presence of beeswax may induce flocculation or coalescence of emulsion droplets during film formation, potentially leading to the formation of cracks or pinholes due to beeswax particles [16]. Moreover, the excessive addition of wax may not augment the water barrier property of the film due to the disruption of the polymer’s continuous phase.

Conversely, the oxygen permeability of PVA/PAA composite films exhibited an increase with escalating beeswax content (Table 1), notably significant when exceeding 5% beeswax content. Similar findings were reported in a previous study [39] involving biopolymer films derived from pea starch, which experienced an increase in oxygen permeability upon excessive beeswax addition. This elevation in oxygen permeability can be attributed to several factors. Firstly, the wax materials within the film matrix can serve as hydrophobic channels for oxygen molecule diffusion. Secondly, the interface between beeswax particles and the polymer could facilitate pathways for oxygen penetration, enhancing its diffusivity. Furthermore, the film’s hydrophobic surface, stemming from increased hydrophobicity due to beeswax addition, accelerates oxygen absorption from the atmosphere, thereby increasing oxygen solubility [16]. Numerous food products, including butter, ramen, and snacks, contain oil and fat. Consequently, the oil barrier property of the film plays a crucial role in preventing external greasiness in fatty food packaging [40]. As depicted in Table 1, the oil permeability of PVA/PAA composite films decreased with escalating beeswax content. Beeswax molecules, forming emulsion droplets within the polymer matrix, contribute to a densely dispersed network. Moreover, considering the melting temperature of beeswax (62–64 °C) lower than the oil permeation test temperature (25 °C), the beeswax molecules in the film matrix may transform into fat crystals enveloped by polymer molecules during film formation, enhancing oil resistance. Additionally, findings from Dang et al. [41] suggest that oil permeability of biopolymer films decreased with the dispersion of nanoparticles within the film matrix.

### 3.8. Mechanical Property of the Film

The tensile strength and elongation at break results of the PVA/PAA composite films with varying beeswax content are presented in Table 1. The tensile strength of the control film (without beeswax) measured 19.71 ± 1.39 MPa. According to a previous study, the mechanical property of PVA film could be enhanced by PAA that acts as a crosslinking agent [12,42]. In this study, however, the tensile strength of PVA/PAA composite films declined with increasing beeswax content due to the substitution of strong polymer–polymer interactions with weaker polymer–wax interactions [43]. This decrease was more pronounced when incorporating high levels of beeswax (exceeding 10%) into the PVA/PAA composite film, attributed to the inability of excessive beeswax to uniformly disperse within the polymer matrix. Additionally, the agglomeration of beeswax could contribute to crack or pinhole formation, further reducing the film’s tensile strength. The elongation at break of PVA/PAA composite films lacking beeswax measured 274.27 ± 15.57%. This value escalated to 323.33 ± 3.55% in the composite film containing 5% beeswax. This increase is credited to the plasticizing role of beeswax within the polymer matrix. Waxes serve as effective plasticizers in polymer films, augmenting polymer chain mobility and thereby enhancing film elasticity. Similar findings were reported by Rodrigues et al. [13], indicating increased elongation at break of cassava starch films upon the inclusion of carnauba wax. Correspondingly, the addition of beeswax enhanced the elongation at break of the guar gum film [44]. However, excessive incorporation of beeswax in the PVA/PAA composite film may disrupt the film structure and compromise its stretchability. Consequently, the elongation at break of the PVA/PAA composite films declined with an increase in beeswax content from 5% to 20%.

## 4. Conclusions

This study explored the impact of beeswax content on the emulsion stability of PVA/PAA film-forming solutions, as well as the physical, barrier, and mechanical properties of composite films. PVA functioned both as a surfactant, stabilizing beeswax, and as a polymer matrix for the composite film. The introduction of beeswax into the PVA and PAA mixture solution facilitated the formation of emulsion droplets. However, excessive beeswax led to emulsion droplet agglomeration due to insufficient surfactant molecules, resulting in the formation of cream layers atop the film-forming solution. This emulsion destabilization during film formation caused defects in the dried films. Morphological investigations revealed uneven and irregularly distributed beeswax particles on the surface of the composite film containing 20% beeswax. Moreover, the incorporation of beeswax into the polymer matrix led to reduced water absorption, solubility in water, and WVP of the films. However, there was an observed increase in oxygen permeability with rising beeswax content. Additionally, the tensile strength and elongation at break of the films notably declined upon incorporating 15% or more beeswax. Therefore, it is evident that an optimal level of hydrophobic compounds, including beeswax, along with their homogeneous distribution, plays a crucial role in enhancing various film properties such as barrier and mechanical properties.

## Figures and Tables

**Figure 1 polymers-16-00310-f001:**
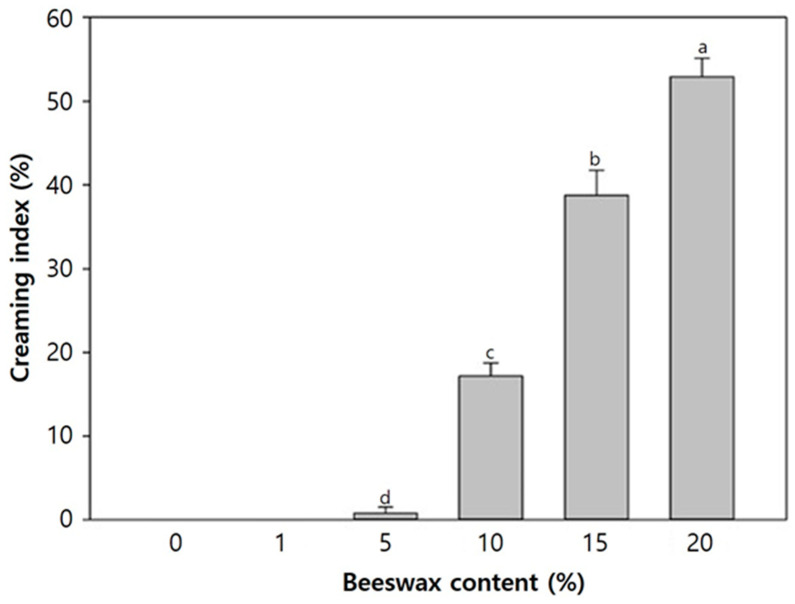
Creaming index of PVA/PAA composite film-forming solutions with different beeswax content. Mean values with different letters are significantly different (*p* < 0.05) by independent sample *t*-test, n = 3.

**Figure 2 polymers-16-00310-f002:**
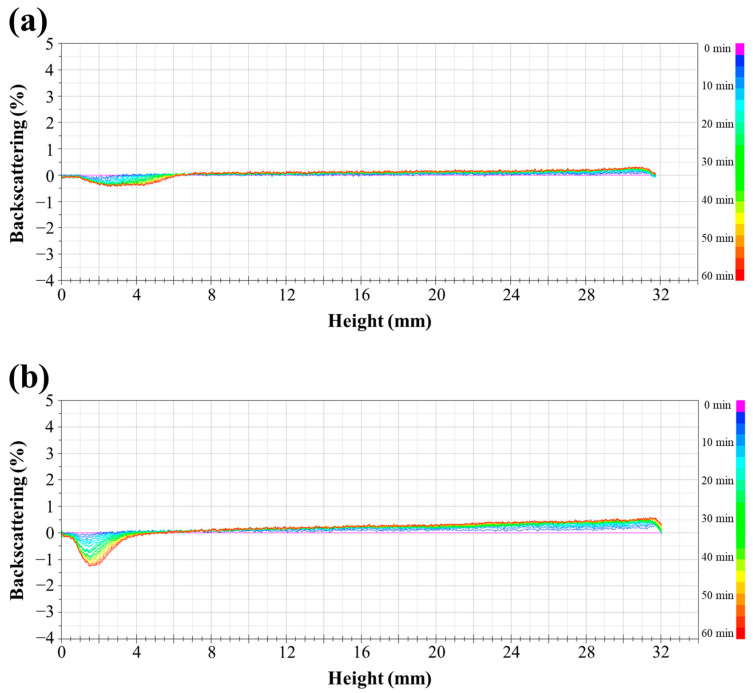
Backscattering profiles of PVA/PAA composite film-forming solutions with different beeswax content: (**a**) 1%, (**b**) 5%, (**c**) 10%, (**d**) 15%, and (**e**) 20%.

**Figure 3 polymers-16-00310-f003:**
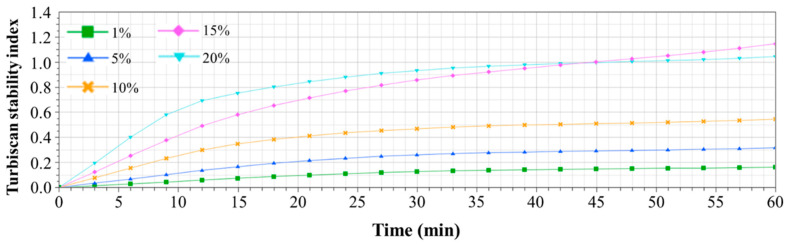
Turbiscan stability index (TSI) values of PVA/PAA composite film-forming solutions with different beeswax content.

**Figure 4 polymers-16-00310-f004:**
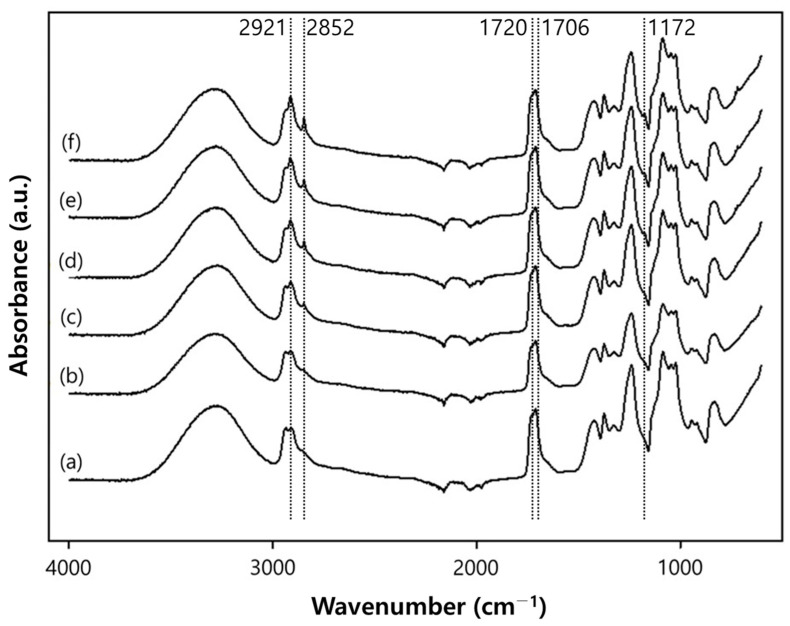
FTIR spectra of PVA/PAA composite films with different beeswax content: (**a**) 0%, (**b**) 1%, (**c**) 5%, (**d**) 10%, (**e**) 15%, and (**f**) 20%.

**Figure 5 polymers-16-00310-f005:**
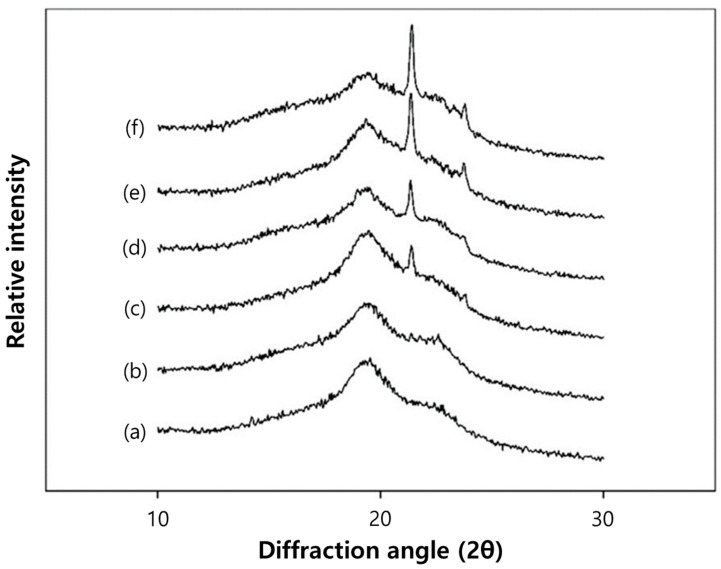
XRD spectra of PVA/PAA composite films with different beeswax content: (**a**) 0%, (**b**) 1%, (**c**) 5%, (**d**) 10%, (**e**) 15%, and (**f**) 20%.

**Figure 6 polymers-16-00310-f006:**
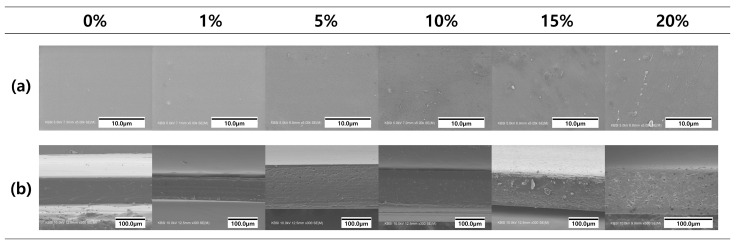
FE-SEM micrographs of (**a**) surface and (**b**) cross-section of PVA/PAA composite films with different beeswax content.

**Figure 7 polymers-16-00310-f007:**
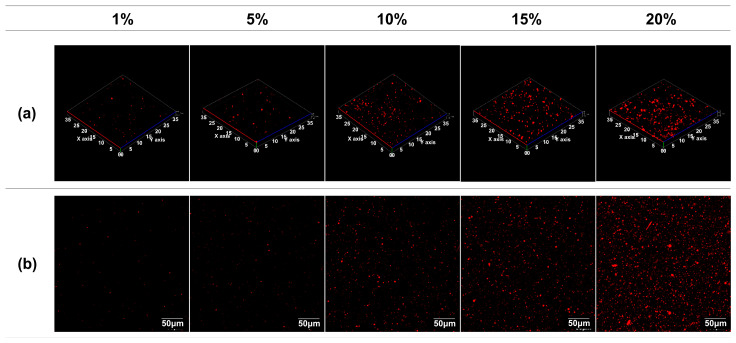
Confocal micrographs of PVA/PAA composite films with different beeswax content: (**a**) 3D images and (**b**) 2D images.

**Table 1 polymers-16-00310-t001:** Physical, barrier, and mechanical properties of PVA/PAA composite films incorporated with beeswax ^(1)^.

% Beeswax	Water Absorption (%)	Solubility (%)	Water Vapor Permeability 10^−10^ (g Pa^−1^ s^−1^ m^−1^)	Oxygen Permeability (cc m^−1^ day^−1^ atm^−1^)	Oil Permeability (g mm m^−2^ day^−1^)	Tensile Strength (MPa)	Elongation at Break (%)
0	11.91 ± 0.11 ^a^	23.98 ± 0.23 ^a^	3.20 ± 0.21 ^a^	1.15 ± 0.49 ^a^	0.177 ± 0.011 ^a^	19.71 ± 1.39 ^a^	274.27 ± 15.57 ^bc^
1	11.51 ± 0.19 ^a^	21.14 ± 1.12 ^b^	3.04 ± 0.3 ^a^	1.15 ± 0.07 ^a^	0.161 ± 0.002 ^ab^	18.91 ± 1.66 ^ab^	301.0 ± 10.01 ^ab^
5	11.40 ± 0.48 ^a^	16.84 ± 0.05 ^c^	2.41 ± 0.14 ^b^	2.6 ± 0.14 ^b^	0.156 ± 0.002 ^ab^	17.51 ± 0.54 ^b^	323.33 ± 3.55 ^a^
10	10.30 ± 0.1 ^b^	11.67 ± 0.1 ^d^	2.24 ± 0.12 ^b^	2.7 ± 0.28 ^b^	0.129 ± 0.028 ^bc^	14.39 ± 0.36 ^c^	247.10 ± 44.66 ^c^
15	10.13 ± 0.6 ^b^	9.67 ± 0.08 ^e^	1.61 ± 0.13 ^c^	3.3 ± 0.28 ^bc^	0.083 ± 0.0029 ^d^	6.90 ± 0.98 ^d^	98.90 ± 10.35 ^d^
20	8.98 ± 0.61 ^c^	7.90 ± 0.03 ^f^	1.64 ± 0.13 ^c^	3.75 ± 0.21 ^c^	0.096 ± 0.041 ^cd^	6.73 ± 0.79 ^d^	97.34 ± 8.21 ^d^

^(1)^ Mean values with different letters within a given column are significantly different (*p* < 0.05) by independent sample *t*-test, n = 3.

## Data Availability

Data are contained within the article.

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
