# Peer review of "Characterization of Polyvinyl Alcohol (PVA)/Polyacrylic Acid (PAA) Composite Film-Forming Solutions and Resulting Films as Affected by Beeswax Content"

_polymers, 2024, doi:10.3390/polym16030310_

Round 1
Reviewer 1 Report
Comments and Suggestions for Authors
The study on the characterization of polyvinyl alcohol (PVA)/polyacrylic acid (PAA) composite film with beeswax content is commendable for its contribution to eco-friendly and sustainable food packaging solutions. The innovative approach of incorporating beeswax into the PVA/PAA composite film is particularly noteworthy, as it significantly improves the barrier properties against oxygen, water, and oil, which are crucial factors in food packaging.
The research provides valuable insights into the impact of varying beeswax content on the physical, structural, and barrier properties of the film. It's impressive how the study addresses the challenges associated with high beeswax content, such as phase separation and destabilization, by employing methods like turbidimetric analysis, scanning electron microscopy (SEM), and confocal laser scanning microscopy (CLSM). These analyses help in understanding the non-homogeneous structures formed due to beeswax agglomeration and their implications on the film's properties.
Moreover, the evaluation of the mechanical properties of the films, including tensile strength and elongation, is crucial for determining the optimal beeswax content. The choice of a 10% beeswax content for the PVA/PAA composite film, based on a balance between barrier properties and mechanical strength, demonstrates a thoughtful approach to material optimization.
In summary, this study represents a significant step forward in the development of environmentally friendly food packaging materials. The PVA/PAA composite film with 10% beeswax, with its enhanced barrier properties and potential for commercial use, offers a promising solution for the food packaging industry's move towards sustainability.
I have only few suggestions to improve the manuscript:
1) Abbreviations should be explained for the first time in text of manuscript apart from abstract (example CLSM)
2) Figure 1 add unities to axis x
3) In materials add CAS numbers of chemicals.
4) In materials add which kind of water did you use and add name of company of distilled water apparatus.
5) Add origin of used beeswax.
Reviewer 2 Report
Comments and Suggestions for Authors
The issue of the manuscript is relevant and of interest from a practical point of view. The application of biofilms, as in this manuscript, attracts attention, considering the importance of improving the properties of the films. This manuscript meets the criteria for publication. There are some minor issues listed as follows:
1. In Figure 1, the labels that appear as a, b, c, and d are missing and considered to be included either in the inset of the graph or in the figure caption.
2. What is the explanation for TSI showing a higher value after 45 min for the sample with beeswax of 15% compared to the sample with 20%
3. Use guidance (lines) in Figure 4 to explain the FTIR results for the reader’s help.
4. The scale bar of micrographs in Figure 6 is not legible; consider improvement.
